# A DFT Study of Hydrogen Storage in High-Entropy Alloy TiZrHfScMo

**DOI:** 10.3390/nano9030461

**Published:** 2019-03-20

**Authors:** Jutao Hu, Huahai Shen, Ming Jiang, Hengfeng Gong, Haiyan Xiao, Zijiang Liu, Guangai Sun, Xiaotao Zu

**Affiliations:** 1School of Physics, University of Electronic Science and Technology of China, Chengdu 610054, China; hujutao_uestc@sina.com (J.H.); mjianglw@gmail.com (M.J.); xtzu@uestc.edu.cn (X.Z.); 2Institute of Nuclear Physics and Chemistry, China Academy of Engineering Physics, Mianyang 621900, China; huahaishen@caep.cn (H.S.); guangaisun@163.com (G.S.); 3Department of ATF R & D, China Nuclear Power Technology Research Institute Co., Ltd., Shenzhen 518000, China; gonghengfeng@cgnpc.com.cn; 4Department of Physics, Lanzhou City University, Lanzhou 730070, China

**Keywords:** DFT, high-entropy alloys, hydrogenation, projected density of state, overlap population

## Abstract

In recent years, high-entropy alloys have been proposed as potential hydrogen storage materials. Despite a number of experimental efforts, there is a lack of theoretical understanding regarding the hydrogen absorption behavior of high-entropy alloys. In this work, the hydrogen storage properties of a new TiZrHfScMo high-entropy alloy are investigated. This material is synthesized successfully, and its structure is characterized as body-centered cubic. Based on density functional theory, the lattice constant, formation enthalpy, binding energy, and electronic properties of hydrogenated TiZrHfScMo are all calculated. The calculations reveal that the process of hydrogenation is an exothermic process, and the bonding between the hydrogen and metal elements are of covalent character. In the hydrogenated TiZrHfScMo, the Ti and Sc atoms lose electrons and Mo atoms gain electrons. As the H content increases, the <Ti–H> bonding is weakened, and the <Hf–H> and <Mo–H> bonding are strengthened. Our calculations demonstrate that the TiZrHfScMo high-entropy alloy is a promising hydrogen storage material, and different alloy elements play different roles in the hydrogen absorption process.

## 1. Introduction

The accelerating global anxiety about the energy crisis, environmental pollution, and climate change has spurred interest in finding alternative and environmentally friendly energy technologies and resources [1]. As one of the hot research topics in the field of energy in recent years, hydrogen energy can provide a renewable and sustainable solution for improving energy utilization efficiency and alleviating environmental pollution. However, hydrogen storage is a challenging task [2]. Among the various storage materials, alloys and intermetallic compounds such as FeTi [3], Mg_3_TiNi_2_ [4], and Zr(Cr_0.5_Ni_0.5_)_2_ [5] have been taken as promising materials for hydrogen storage due to their high volume density, safety, and reversibility [6]. In particular, high-entropy alloys (HEAs) with a body-centered cubic (BCC) structure have become one of the most developed groups of new materials [7].

HEAs, first proposed by Yeh et al. [8], are composed of five or more principal elements in equimolar ratios or varying from 5 to 35 at.%. HEAs usually form simple solid solutions with BCC or face-centered cubic (FCC) structures, and exhibit good mechanical properties (e.g., high hardness and good plasticity). Zhang et al. [9] demonstrated that the atomic radius differences and the absolute mixing entropy affect the formation of solid solutions for multi-component alloys. They found that the entropy of mixing should be in the range of 12–17.5 JK^−1^mol^−1^ and the atomic radius differences (*θ*) should be smaller than 6.6%. Here, *θ* is defined as:$$\theta =\sqrt{\overset{n}{\underset{i=1}{\sum }}C_{i}\cdot {\left(1-\frac{r_{i}}{\bar{r}}\right)}^{2},}$$
where *n* is the number of components of the alloy, *r_i_* is the atomic radius of element *i*, *C_i_* is the percentage of each component of the alloy, and $\bar{r}\left(\sum_{i=1}^{n}C_{i}\cdot r_{i}\right)$ is the average atomic radius. The large lattice distorsion for high *θ* value (>6.6%) may lead to the formation of intermetallic precipitations, such as Laves phases [10,11].

In recent years, the potential of HEAs as hydrogen storage materials has attracted more and more attention. Kunce et al. [12] investigated the hydrogen storage ability of TiZrVCrFeNi and found that its maximum hydrogen capacity is 1.81 wt.% at 100 bar and 50 °C after activation for 2 h at 500 °C under vacuum, and 1.56 wt.% after additional annealing at 1000 °C for 24 h. Sahlberg et al. [13] reported that the TiZrHfVNb HEA with BCC structure can absorb hydrogen readily with a plateau pressure of 0.1 bar (H_2_) at 299 °C and the maximum measured hydrogen storage capacity is 2.7 wt.%. Karlsson et al. [14] studied the hydrogenation mechanism of the TiZrHfVNb HEA under different pressure and temperature conditions, and found that the large lattice distortions caused by atomic radius difference in the HEA are favorable for absorption in both octahedral and tetrahedral sites.

Apart from the above experimental studies, few theoretical investigations of hydrogen storage in high-entropy alloys have been reported, since it is difficult to model HEAs because the alloy elements are randomly distributed in the lattice sites. It remains unknown how the hydrogen interacts with the HEA and how the hydrogen absorption affects the geometrical structure, binding energy, charge distribution, overlap population, etc. In this study, we first synthesize a novel HEA (i.e., TiZrHfScMo with BCC structure) by the arc-melting method. The atomic-size difference (*θ*), the enthalpy of mixing (*ΔH_mix_*), and the entropy of mixing (*ΔS_mix_*) for TiZrHfScMo alloy were 4.87%, 2.24 KJ/mol, and 13.4 J/(K·mol), respectively, satisfying the criteria for HEA proposed by Zhang et al. [11]. It is expected that this HEA could favor efficient hydrogen storage. A density functional theory (DFT) method was used to investigate the structural and electronic properties of hydrogenated TiZrHfScMo with different hydrogen concentrations and evaluate its potential as a hydrogen storage material. The present study may open up new possibilities for hydrogen storage based on this HEA and promote further theoretical and experimental investigations of the related topic.

## 2. Computational and Experimental Details

The first-principles calculations are conducted using the Vienna Ab initio Simulation Package (VASP) code (Vienna, Austria) [15]. The projector augmented-wave (PAW) method [16,17] is employed to treat the interaction between ions and electrons. The exchange-correlation effects are treated within the framework of the local density approximation (LDA). The electronic configurations for the PAW potentials are 3d^3^ 4s^1^ for Ti, 4s^2^4p^6^4d^3^5s^1^ for Zr, 5d^3^6s^1^ for Hf, 3d^2^ 4s^1^ for Sc, 4d^5^5s^1^ for Mo, and 1s^1^ for H. The cutoff energy of the plane waves is 650 eV and the Brillouin zone integration is performed by the Monkhorst–Pack scheme with a 2 × 2 × 2 k-point mesh. For the BCC TiZrHfScMo HEA, we employ a 5 × 5 × 2 supercell containing 100 atoms. Eleven compositions are considered for hydrogenated HEA (i.e., TiZrHfScMo-H*_x_*), with *x* varying from 0.25 to 10. For each composition, we generate 100 different configurations using the Python program and determine the most energetically stable configuration by structural optimization. In the total energy calculations, the energy and force convergence limits are 1 × 10^−4^ eV/atom and −1 × 10^−3^ eV/Å, respectively. 

Experimentally, the equiatomic Ti, Zr, Hf, Sc, and Mo metals are weighed to synthesize the TiZrHfScMo alloy by the arc-melting method. All of the raw materials are in the shape of small particles between 1 and 10 mm and had a purity of 99.99%. The alloy ingot is re-melted five times to improve the homogeneity of each element. X-ray diffraction (XRD) characterization is performed on X’Pert PRO MPD equipment (PANalytical B.V., Almelo, The Netherlands) working at 45 kV and 40 mA to determine the crystal structure of as-obtained TiZrHfScMo alloy. The grain size of the TiZrHfScMo alloy is examined by Electron Backscatter Diffraction (EBSD) using an Oxford Nordlys Max^2^ in the Zeiss Auriga workstation (Jena, Germany). The samples for EBSD measurements are carefully prepared by mechanical grinding using diamond abrasive paper and subsequent final polishing using a vibratory polisher. 

## 3. Results and Discussion

### 3.1. The Structural Parameters of the TiZrHfScMo HEA before and after Hydrogenation

Figure 1a presents the XRD of the as-obtained TiZrHfScMo alloy. The XRD pattern containes four sharp diffraction peaks, which is refined by Rietveld analysis based on the available crystal ZrNb alloy that has BCC structure and the lattice constant of 3.439 Å [18]. It is confirmed that the structure of TiZrHfScMo alloy is in good agreement with the ZrNb alloy and has a lattice constant of 3.444 Å. In comparison, the simulated pattern of TiZrHfScMo alloy is also shown in the XRD pattern. The nominal composition of the alloy is measured by energy-dispersive spectrometry (EDS) as Ti_0.20_Zr_0.18_Hf_0.21_Sc_0.21_Mo_0.20_, which is consistent with the designed equiatomic TiZrHfScMo alloy. Figure 1b shows the EBSD map of TiZrHfScMo alloy, and the grain size of this alloy is measured as 104.2 ± 52.3 um, indicating that the TiZrHfScMo alloy has a great crystallinity. The above results indicate that a single phase of TiZrHfScMo high-entropy alloy is synthesized successfully. 

Theoretically, the lattice constant of TiZrHfScMo HEA is determined to be 3.325 Å, deviating from the experimental value by 3.45%, which may be a result of the pseudopotential approximation [19] employed in the calculations. Based on this structure, the hydrogen atoms are absorbed into the crystals in an accumulative and random way, with H concentration varying from 0.05 wt.% to 2.14 wt.%, and a full structural relaxation is then performed for these HEA hydrides. Figure 2 presents a schematic view of the optimized TiZrHfScMo and TiZrHrScMo-H_x_. As seen in Figure 2, the displacement of alloy elements is more significant with the increasing hydrogen content. Note that no intermetallic compounds are formed in the TiZrHfScMo hydrides. The calculated lattice constants are summarized in Table 1, along with the available experimental results for comparison. The variation of the lattice constant for TiZrHfScMo-H_x_ is illustrated in Figure 3. The lattice constant increased with the increasing hydrogen content, which results in volume expansion varying from 0.18% to 7.07%. 

### 3.2. The Energetic Properties of the TiZrHfScMo HEA and Its Hydrides

To evaluate the stability of the HEA hydrides, the formation enthalpy (*H_form_*), defined as the energy difference between the final compound and initial constituents, is calculated. The *H_form_* is calculated by the following equation [20]: $$H_{form}=\left[E_{tot}\left(MH_{x}\right)-E\left(Ti\right)-E\left(Zr\right)-E\left(Hf\right)-E\left(Sc\right)-E\left(Mo\right)-\frac{x}{2}E\left(H_{2}\right)\right]/N,$$
where *M* is the HEA TiZrHfScMo, *x* is the number of hydrogen atoms per formula unit, and *N* is the number of atoms in the unit cell. In the equation, $E_{tot}\left(MH_{x}\right)$ refers to the total energy of the hydrides, $E\left(H_{2}\right)$ is the total energy of the H_2_ molecule, and E(Ti), E(Zr), E(Hf), E(Sc) and E(Mo) are the single atomic energies of the pure elements in their respective stable solid states. The calculated *H_form_* is summarized in Table 2 and plotted in Figure 4. As shown in Table 2, the *H_form_* of the TiZrHfScMo-H*_x_* are all negative, suggesting that the process of hydrogenation is an exothermic process and the hydrides with H concentration varying from 0.05 wt.% to 2.14 wt.% are all energetically stable [20]. These results indicate that the TiZrHfScMo HEA has great potential to be a hydrogen storage material [21]. As shown in Figure 4, the *H_form_* decreases significantly when the H content increases to 1.72 wt.% and then the formation enthalpy started to increase, suggesting that the hydride formation becomes relatively more difficult when the H content reached a certain level. The decreased formation enthalpy may be due to the lattice expansion, which leads to an increasing volume of the HEA TiZrHfScMo and makes it easier for hydrogen atoms to occupy the sites [22]. 

The hydrogen binding energy (*E_B_*) is defined as [23,24]: $$E_{B\;}=\frac{1}{x}\left[E_{tot}\left(M\right)+\frac{x}{2}E\left(H_{2}\right)-E_{tot}\left(MH_{x}\right)\right].$$

This is also calculated to investigate the interaction between the hydrogen and HEA in the hydrides. Here, $E_{tot}\left(MH_{x}\right)$ is the total energy of the TiZrHfScMo with *x* hydrogen atoms, $E_{tot}\left(M\right)$ is the total energy of the HEA without hydrogen, and $E\left(H_{2}\right)$ is the total energy of the hydrogen molecule. The calculated binding energies are given in Table 2, and Figure 4 presents the variation of binding energy for TiZrHfScMo-H*_x_* with hydrogen content. Note that the *E_B_* is large enough, suggesting that the process of hydrogen absorption is chemical sorption, and there may be covalent bonding between the hydrogen and metal elements [2]. Furthermore, the decline of *E_B_* indicates that the stability of the system is gradually decreasing [23]. Therefore, if the concentration of hydrogen reaches a certain level, the hydride will not exist stably any more.

### 3.3. Electronic Structures and Mulliken Population Analysis of TiZrHfScMo-H_x_

To explore how the hydrogen occupation influences the electronic structure and charge distribution of the TiZrHfScMo HEA, the density of state distribution, Mulliken charge, and overlap population were analyzed. The projected density of state (PDOS) of TiZrHfScMo with spin-up and spin-down channels is illustrated in Figure 5, along with those for TiZrHfScMo-H_0.25_, TiZrHfScMo-H_1.5_, and TiZrHfScMo-H_3_ for comparison, which are presented in Figure 6. It is shown that all the structures behaved with metallic characters [5].

For TiZrHfScMo (see Figure 5), the valence bands from −6 to −4.5 eV are mainly contributed by the s orbital of all elements, and the d orbitals of Ti, Zr, Hf, Sc, and Mo dominated in the energy range of −4 to 3 eV. The contribution to the total DOS by the composed components is similar, and there is a strong hybridization between these elements, which may result in high stability of HEAs [8]. As shown in Figure 6, for TiZrHfScMo-H_0.25_, TiZrHfScMo-H_1.5_, and TiZrHfScMo-H_3_, the H 1s orbitals mainly interact with the s orbitals of alloy elements in the energy range of −8 to −4.5 eV, indicative of strong bonding between the H and the metal elements. Especially, the hybridization between the H–s and Hf–s is the most significant, meaning that the <H–Hf> interaction may be stronger than the interaction between the H and other metal elements. These results suggest that the structural stability of hydrides may be influenced by the different bonding between H and metal elements. 

The Mulliken charge, which describes the charge of distribution, is analyzed to gain a fundamental insight into the charge transfer between hydrogen and alloying atoms in the TiZrHfScMo hydrides [25]. The Mulliken charge is calculated by employing the Spanish Initiative for Electronic Simulations with Thousands of Atoms (SIESTA) code [26]. The results are listed in Table 3 and plotted in Figure 7a. In the calculations, the electronic configuration for valence states are 3d^2^4s^2^ for Ti, 4d^2^5s^2^ for Zr, 5d^2^ 6s^2^ for Hf, 3d^1^4s^2^ for Sc, 4d^5^5s^1^ for Mo, and 1s^1^ for H. If the average value of the Mulliken charge is positive, it means that there must be electron loss for this element after bonding, while a negative value implies electron gain [20]. The larger the negative (or positive) value of the charge for an element, the stronger the ability to gain (or lose) electrons. As can be seen from Figure 7a, the Ti and Sc atoms in all cases lose electrons, while the H and Mo atoms gain electrons in all the considered hydrides. The Mo has a stronger ability to gain electrons, and with the increasing of the hydrogen content, the charge values of H and Mo get closer. A possible explanation is that the electronegativity of Mo is the strongest among all the metal elements in the investigated system and is similar to the electronegativity of H [27]. The overlap population is another way to investigate the charge distribution on the chemical bond. A higher positive population indicates a higher covalent nature in the chemical bonding, and a negative value of overlap population is responsible for the antibonding states in the chemical bonding [28]. The average values of the overlap population between the hydrogen and metal elements are listed in Table 4 and plotted in Figure 7b. As shown in Figure 7b, the overlap population between H and metal elements is always positive, indicative of covalent bonding interaction between them [29]. Furthermore, with increase of the hydrogen concentration, the <Ti–H> bonding is weakening, and the <Hf–H> and <Mo–H> bonding become stronger, indicative of different roles of metal atoms during the absorption and desorption processes of hydrogen storage.

## 4. Conclusions

A novel TiZrHfScMo high-entropy alloy is designed, and its hydrogen storage properties are investigated by first-principles calculations. Experimental study shows that this material can be synthesized, and its structure is BCC phase. The structural parameters, energetic properties, and electronic structures of the HEA TiZrHfScMo and its hydrides are investigated by the DFT method. Our calculations show that the process of hydrogenation is an exothermic process and the hydrides with H concentration varying from 0.05 wt.% to 2.14 wt.% are all energetically stable. The formation enthalpies decrease significantly when the H content increase to 1.72 wt.% and then start to increase from H content of 2.14 wt.%. The calculated binding energies suggest that the process of hydrogen absorption is a chemical sorption, and there may be covalent bonding between the hydrogen and metal elements. As the hydrogen content increases, the bonding interaction between the hydrogen and TiZrHfScMo are weakened. A detailed charge analysis show that Ti and Sc atoms lose electrons, and Mo atoms gain electrons in the hydrogenated TiZrHfScMo HEA. The overlap population between H and metal elements indicates that covalent bonding interaction existed between them. As the hydrogen concentration increases, the <Ti–H> bonding is weakened, and the <Hf–H> and <Mo–H> bonding are strengthened, suggesting that Ti, Sc, Hf, and Mo play different roles during the absorption processes of hydrogen storage. Our calculations suggest that the HEA TiZrHfScMo is potentially a good candidate hydrogen storage material.

## Figures and Tables

**Figure 1 nanomaterials-09-00461-f001:**
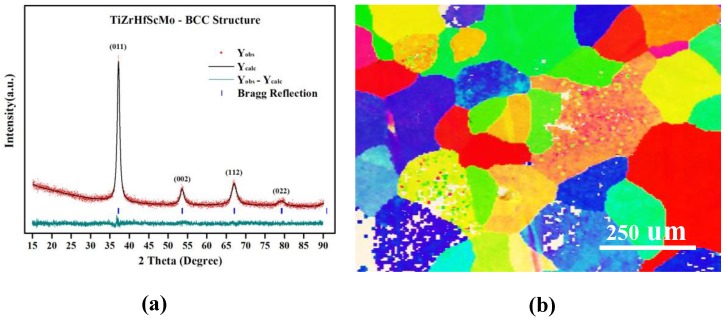
(**a**) The XRD pattern of TiZrHfMoNb alloy; (**b**) The EBSD map of TiZrHfScMo alloy.

**Figure 2 nanomaterials-09-00461-f002:**
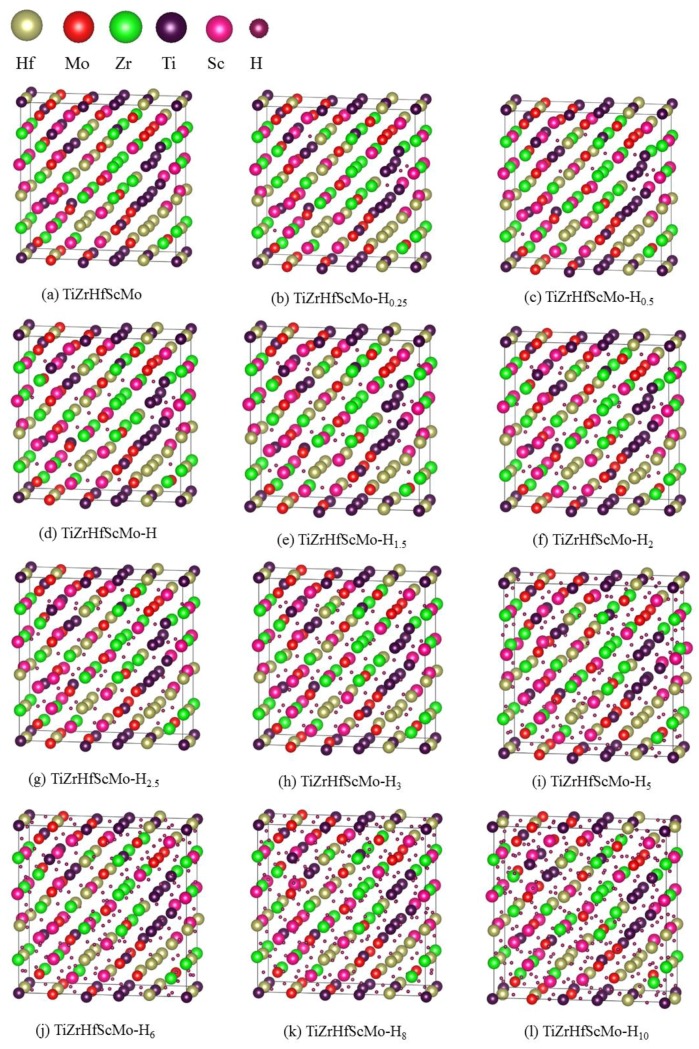
Illustration of schematic views of (**a**) the HEA TiZrHfScMo; (**b**–**l**) TiZrHrScMo-H_x_, where *x* is the number of hydrogen atoms per formula unit. The yellow, red, green, purple, and pink spheres represent Hf, Mo, Sc, Ti, and Zr atoms, respectively. The hydrogen atoms are represented by the smaller wine-colored spheres.

**Figure 3 nanomaterials-09-00461-f003:**
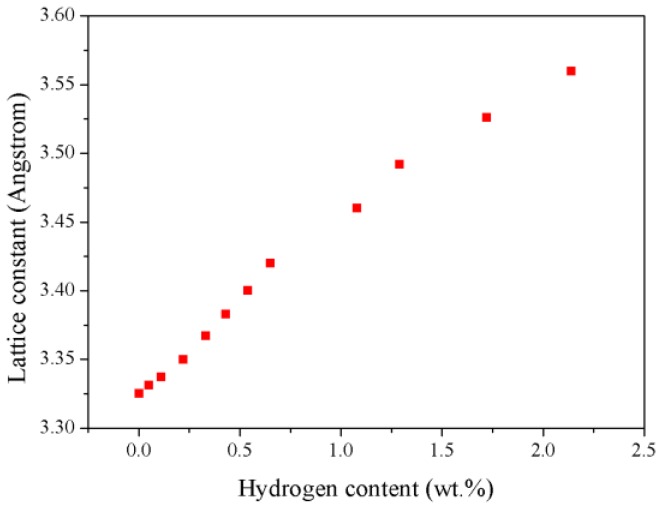
The variation of lattice parameters for the TiZrHfScMo hydrides with increasing hydrogen content.

**Figure 4 nanomaterials-09-00461-f004:**
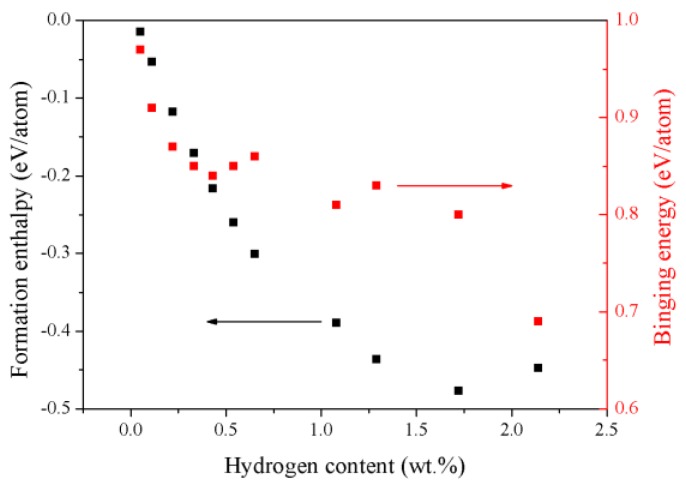
The calculated formation enthalpy and binding energy for the TiZrHfScMo hydrides with different hydrogen content.

**Figure 5 nanomaterials-09-00461-f005:**
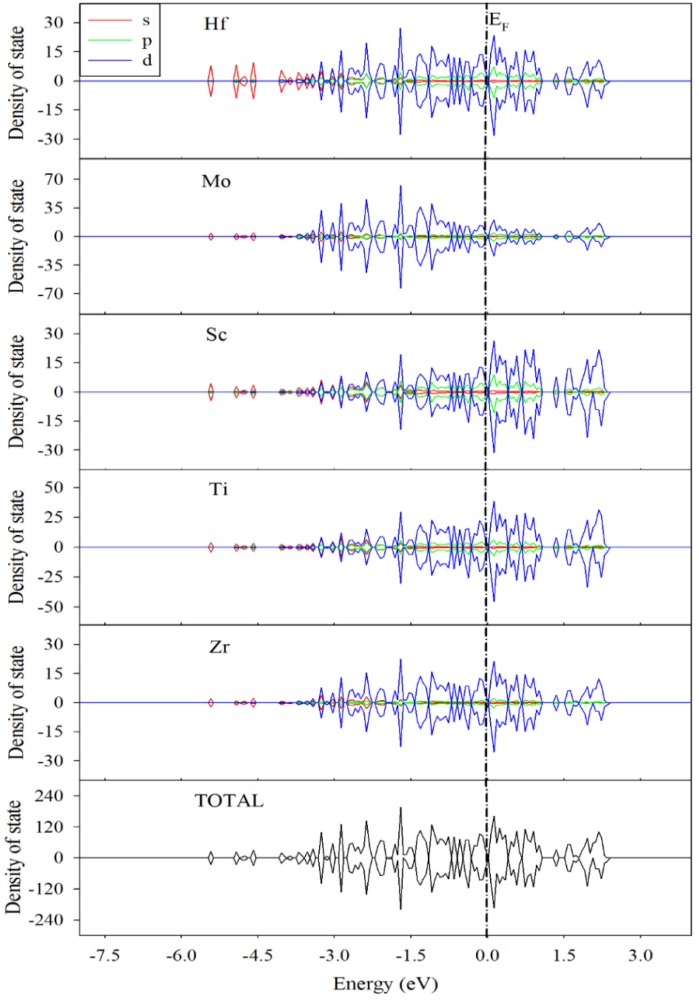
The projected density of state (PDOS) of TiZrHfScMo. The Fermi level was set to zero.

**Figure 6 nanomaterials-09-00461-f006:**
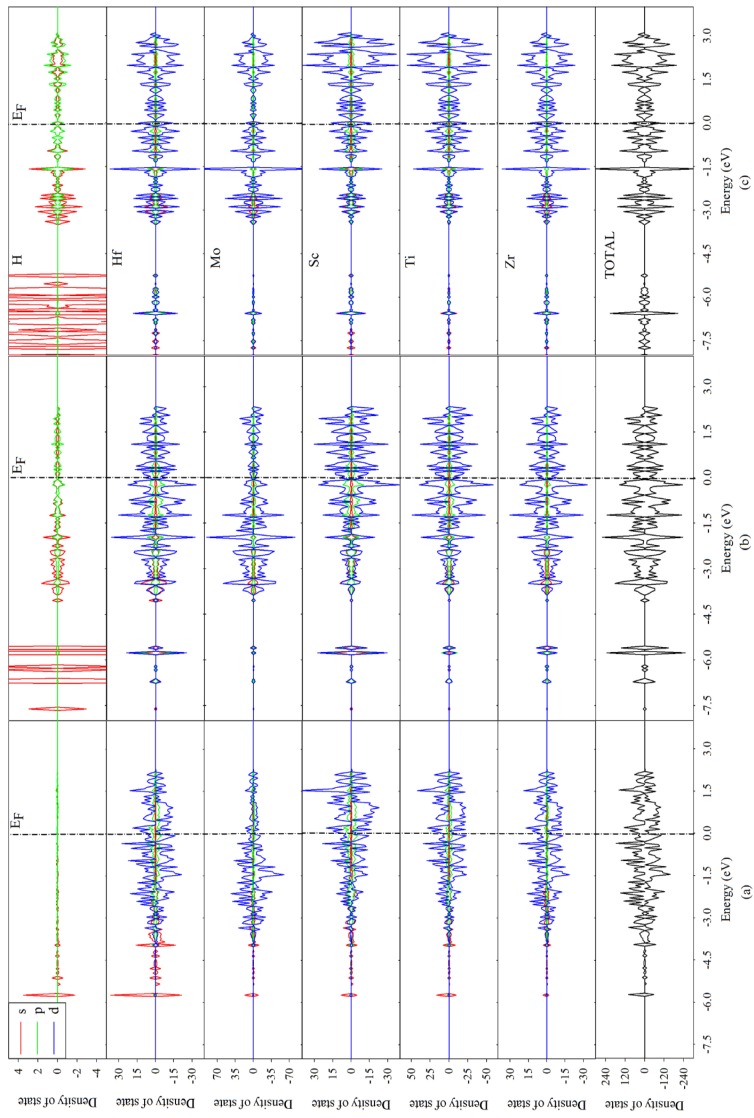
The projected density of state (PDOS) of (**a**) TiZrHfScMo-H_0.25_, (**b**) TiZrHfScMo-H_1.5_, and (**c**) TiZrHfScMo-H_3_. The Fermi level was set to zero.

**Figure 7 nanomaterials-09-00461-f007:**
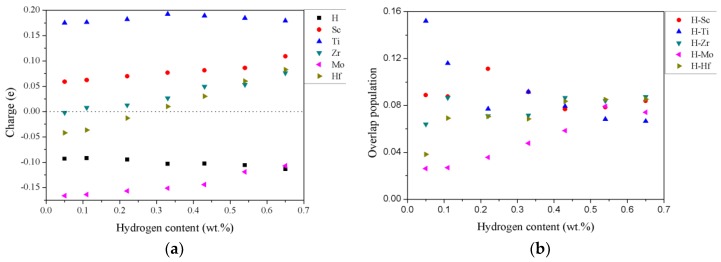
(**a**) The average charge distribution of different elements in the TiZrHfScMo hydrides with increase of the hydrogen content. (**b**) The average overlap population between hydrogen and metal elements in the hydrides of HEA TiZrHfScMo with increase of the hydrogen content.

**Table 1 nanomaterials-09-00461-t001:** The calculated and experimental lattice constants for M–H*_x_* (*M* = TiZrHfScMo, *x* is the number of hydrogen atoms per formula unit).

Config.	Lattice Constant (Å)
Exp.	3.444
M	3.325
M–H_0.25_	3.331
M–H_0.5_	3.337
M–H	3.350
M–H_1.5_	3.367
M–H_2_	3.383
M–H_2.5_	3.400
M–H_3_	3.420
M–H_5_	3.46
M–H_6_	3.492
M–H_8_	3.526
M–H_10_	3.56

**Table 2 nanomaterials-09-00461-t002:** The calculated formation enthalpy and binding energy of the M-H*_x_* (*M* = TiZrHfScMo, where *x* is the number of hydrogen atoms per formula unit).

Config.	Formation Enthalpy (eV/Atom)	Binding Energy (eV/Atom)
M–H_0.25_	−0.0147	0.97
M–H_0.5_	−0.0531	0.91
M–H	−0.1174	0.87
M–H_1.5_	−0.1704	0.85
M–H_2_	−0.2161	0.84
M–H_2.5_	−0.2599	0.85
M–H_3_	−0.3008	0.86
M–H_5_	−0.3894	0.81
M–H_6_	−0.4359	0.83
M–H_8_	−0.477	0.8
M–H_10_	−0.4478	0.69

**Table 3 nanomaterials-09-00461-t003:** The average charge (**e**) of different elements in M–H*_x_* (*M* = TiZrHfScMo, *x* = 0.25, 0.5, 1, 1.5, 2, 2.5, and 3).

Config.	H	Ti	Zr	Hf	Sc	Mo
M–H_0.25_	−0.093	0.175	−0.002	−0.042	0.059	−0.166
M–H_0.5_	−0.092	0.176	0.008	−0.036	0.062	−0.164
M–H	−0.095	0.182	0.013	−0.013	0.070	−0.157
M–H_1.5_	−0.103	0.193	0.026	0.010	0.077	−0.151
M–H_2_	−0.103	0.189	0.049	0.030	0.081	−0.144
M–H_2.5_	−0.106	0.185	0.053	0.060	0.086	−0.119
M–H_3_	−0.113	0.179	0.076	0.083	0.109	−0.107

**Table 4 nanomaterials-09-00461-t004:** The overlap population between hydrogen and metal elements in M–H*_x_* (M = TiZrHfScMo, *x* = 0.25, 0.5, 1, 1.5, 2, 2.5, and 3).

Config.	Ti–H	Zr–H	Hf–H	Sc–H	Mo–H
M–H_0.25_	0.1520	0.0640	0.0382	0.0888	0.0262
M–H_0.5_	0.1159	0.0865	0.0692	0.0875	0.0269
M–H	0.0771	0.0711	0.0706	0.1112	0.0357
M–H_1.5_	0.0917	0.0716	0.0685	0.0915	0.0477
M–H_2_	0.0793	0.0865	0.0835	0.0768	0.0585
M–H_2.5_	0.0683	0.0839	0.0851	0.0784	0.0793
M–H_3_	0.0666	0.0874	0.0853	0.0838	0.0742

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
