# Peer review of "A DFT Study of Hydrogen Storage in High-Entropy Alloy TiZrHfScMo"

_nanomaterials, 2019, doi:10.3390/nano9030461_

Round 1
Reviewer 1 Report
The authors present a high entropy alloy with Density Functional Theory predicted properties with some experimental work. The results indicate the success of a single phase TiZrHfScMo alloy together with enthalpy of formation predictions with H atoms and electronic properties. The results are well explained and presented. The one comment I would make is that since the alloy has also been formed experimentally, that there should be more experimental validation of the results as discussed in the comments below.
· Please include the Rietveld refinement results showing the single phase TiZrHfScMo alloy results with lattice parameters.
· What is the explanation of the deviation between experimental and theoretical results?
· The authors describe in situ XRD however, only one experimental result is given? Was the sample heated and the lattice expansion measured? If so, where are these results and the resulting DFT including the influence of temperature and thermal lattice expansion?
· Was the Pressure-Temperature-Isotherm measured experimentally? If so, where are these results showing the various hydrogen concentrations within the lattice. If not, this should be done to validate the DFT results given.
Author Response
The authors present a high entropy alloy with Density Functional Theory predicted properties with some experimental work. The results indicate the success of a single phase TiZrHfScMo alloy together with enthalpy of formation predictions with H atoms and electronic properties. The results are well explained and presented. The one comment I would make is that since the alloy has also been formed experimentally, that there should be more experimental validation of the results as discussed in the comments below.
Comment 1: Please include the Rietveld refinement results showing the single phase TiZrHfScMo alloy results with lattice parameters:
Response: The synthesized TiZrHfScMo alloy in this study is a new high-entropy alloy which cannot be found in the ICSD database. All of diffraction peaks were carefully indexed as BCC crystal structure, which is very similar to the TiZrHfVNb alloy reported in the reference [Karlsson, D., et al. Inorg. Chem. 2018, 57, 2103-2110].
Comment 2: What is the explanation of the deviation between experimental and theoretical results?
Response: Our calculated lattice constant of 3.325 Å is 3.45 % smaller than the experimental value of 3.444 Å. In DFT calculations, the interaction between ions and electrons is described by pseudopotentials, i.e., the core electrons are frozen and only valence electrons are considered. In some cases, this pseudo-potential approximation results in small difference between theoretical and experimental results. Corresponding explanations have been made in page 3, lines 105-107 in the revised manuscript.
Comment 3: The authors describe in situ XRD however, only one experimental result is given? Was the sample heated and the lattice expansion measured? If so, where are these results and the resulting DFT including the influence of temperature and thermal lattice expansion? Response: The aim of this study is to evaluate the potential of TiZrHfScMo high entropy alloy (HEA) as a hydrogen storage material by density functional theory calculations. This new material has not been synthesized and reported in the literature thus far. Thus, we synthesized the TiZrHfScMo HEA and characterized its structure by in situ XRD. The experimental results in this study proves that the new TiZrHfScMo HEA can be synthesized and its structure is BCC phase. In the future work, the hydrogen storage stability will be investigated experimentally, and a separate publication will be made.
Comment 4: Was the Pressure-Temperature-Isotherm measured experimentally? If so, where are these results showing the various hydrogen concentrations within the lattice. If not, this should be done to validate the DFT results given.
Response: The aim of this study is to evaluate the potential of TiZrHfScMo high entropy alloy (HEA) as a hydrogen storage material by density functional theory calculations. Experimental investigation of the hydrogen storage stability will be carried out in the future work and another publication will present all the results in details.

Reviewer 2 Report
In the work with title "A DFT study of hydrogen storage in high entropy alloy TiZrHfScMo" a theoretical study on the behaviour of TiZrHfScMo alloy in the hydrogen adsorption is carried out. DFT calculations are very attractive and powerful tool and any work in this field is welcome. The present work can be accepted with some minor modifications.
Following, there are some required clarifications:
Is the TiZrHfScMo alloy actually synthesised? How?
Regarding to X-ray diffraction authors reported the lattice constant but not any other parameter such as particle size? Why is important to determine such parameter?
In figure 2 is quite difficult to appreciate differences, apart from the increasing number of hydrogen atoms. Is there any other noticeable difference?
Author Response
In the work with title "A DFT study of hydrogen storage in high entropy alloy TiZrHfScMo" a theoretical study on the behaviour of TiZrHfScMo alloy in the hydrogen adsorption is carried out. DFT calculations are very attractive and powerful tool and any work in this field is welcome. The present work can be accepted with some minor modifications. Following, there are some required clarifications:
Comment 1: Is the TiZrHfScMo alloy actually synthesised? How?
Response: Yes, the equiatomic Ti, Zr, Hf, Sc and Mo metals are weighed to synthesize the TiZrHfScMo alloy by the arc-melting method. All of the raw materials are in the shape of small particles between 1 and 10 mm and have a purity of 99.99%. The alloy ingot is re-melted five times to improve the homogeneity of each element. The details have been described on page 3, lines 89-95.
Comment 2: Regarding to X-ray diffraction authors reported the lattice constant but not any other parameter such as particle size?
Response: The particle size is 104.2±52.3 um, which has been described on page 3, line 92. Comment 3: Why is important to determine the lattice constant?
Response: The TiZrHfScMo alloy in this study is a new high-entropy alloy, for which the structural phase and lattice constant has not been reported thus far and the related information are not available in the literature. In our DFT calculation, the structural phase is necessary for the building of structural model and the lattice constant is necessary to test the validity of our calculations. Hence, the lattice constant is an important parameter.
Comment 4: In figure 2 is quite difficult to appreciate differences, apart from the increasing number of hydrogen atoms. Is there any other noticeable difference
Response: With the increasing number of hydrogen atoms, the displacement of alloy elements is more significant. This is caused by the interaction between hydrogen atoms and metal atoms.

Reviewer 3 Report
The manuscript proposes valuable theoretical insights of the hydrogen absorption behavior of high entropy alloys (TiZrHfScMo).
Despite of all benefits the manuscript must be improved in order to be considered for publishing:
As proposed manuscript is based mostly on theoretical calculations it remains unclear why authors included information on alloy synthesis in "2. Computational details" and Fig.1 and Fig.3 in "3.1. The structural parameters....". Fig. 1 presents the XRD of the alloy as obtained. What is the added value of this XRD results? If it could be keeped would be useful to add XRD results after alloy hydrogenation too.
Both of Fig.1. and Fig. 3 could be excluded or authors need to rewrite Abstract and Conclusions where these experimental activities and results not mentioned.
I would suggest to extend all experimental activities in separate publication and present all experimental results in details.
Author Response
The manuscript proposes valuable theoretical insights of the hydrogen absorption behaviour of high entropy alloys (TiZrHfScMo). Despite of all benefits the manuscript must be improved in order to be considered for publishing:
Comment 1: As proposed manuscript is based mostly on theoretical calculations it remains unclear why authors included information on alloy synthesis in "2. Computational details" and Fig.1 and Fig.3 in "3.1. The structural parameters....".
Response: The aim of this study is to evaluate the potential of TiZrHfScMo high entropy alloy (HEA) as a hydrogen storage material by density functional theory calculations. This new material has not been synthesized and reported in the literature thus far. Thus, we synthesized the TiZrHfScMo HEA and characterized its structure by in situ XRD. The experimental results in this study proves that the new TiZrHfScMo HEA can be synthesized and its structure is BCC phase. In the revised manuscript, the title of section 2 has been modified as “Computational and experimental details”. Comment 2: Fig. 1 presents the XRD of the alloy as obtained. What is the added value of this XRD results? If it could be keeped would be useful to add XRD results after alloy hydrogenation too. Response: Please see the response to comment 1.
Comment 3: I would suggest to extend all experimental activities in separate publication and present all experimental results in details.
Response: Thanks for the reviewer’s suggestions. Yes, experimental investigation of the hydrogen storage stability will be carried out in the future work and another publication will present all the experimental results in details.
Comment 4: Both of Fig.1. and Fig. 3 could be excluded or authors need to rewrite Abstract and Conclusions where these experimental activities and results not mentioned.
Response: In the revised manuscript, the abstract and conclusions have been rewritten.

Round 2
Reviewer 1 Report
The response regarding the experimental work is not yet satisfactory. Simulation and modelling should always be validated were possible with experimental work. Since the alloy has already been synthesised, I would suggest that this experimental work be included in this paper not written as a separate paper.
It appears that the meaning of in situ XRD is not correct in this study. This implies that the sample was heated over time and XRD measurements taken. This does not appear to be the case. Where are the in situ XRD results? Therefore, please review your manuscript accordingly.
With regards to the Rietveld refinement, I assume this was done to calculate the lattice parameters. However, the authors have again, not included this in the revised version. If the structure has already been indexed, it is still possible to model the results with regards to whole pattern fitting in order to determine the lattice parameters. Is this not the case? If so, how was the experimental XRD results determined?
Publication is recommended once these issues are addressed.
Author Response
The response regarding the experimental work is not yet satisfactory. Simulation and modelling should always be validated were possible with experimental work. Since the alloy has already been synthesised, I would suggest that this experimental work be included in this paper not written as a separate paper.
Response:
We now only synthesized successfully the TiZrHfVNb high entropy alloy (HEA) and characterized its structure. Since the TiZrHfVNb HEA is a new material, we need to prove in our theoretical simulations that this material can be synthesized. Also, the structure phase information is necessary for building the structural model in simulation and the lattice parameter is necessary to validate the calculation results.
Due to the limitation of experimental conditions, we don’t have more data except XRD.
Comment 1: It appears that the meaning of in situ XRD is not correct in this study. This implies that the sample was heated over time and XRD measurements taken. This does not appear to be the case. Where are the in situ XRD results? Therefore, please review your manuscript accordingly.
Response: We are sorry for the typo. It should be XRD instead of in situ XRD. In the revised manuscript, the mistake has been corrected.
Comment 2: With regards to the Rietveld refinement, I assume this was done to calculate the lattice parameters. However, the authors have again, not included this in the revised version. If the structure has already been indexed, it is still possible to model the results with regards to whole pattern fitting in order to determine the lattice parameters. Is this not the case? If so, how was the experimental XRD results determined?
Response: The XRD pattern was refined based on the available crystal ZrNb alloy in the literature [ICSD #105261] that has BCC structure and the lattice constant of 3.439 Å. It is confirmed that the structure of TiZrHfScMo alloy is in good agreement with the ZrNb alloy and has a lattice constant of 3.444 Å. In comparison, the simulated pattern of ZrNb alloy is also shown in the XRD pattern.
Corresponding modification has been made on page 3, lines 102-111.
Reviewer 3 Report
-
Author Response
In the revised manuscript, the English language has been improved. Please see the highlited parts in the mansucript.